# A Method of Merging Maps for MUAVs Based on an Improved Genetic Algorithm

**DOI:** 10.3390/s23010447

**Published:** 2023-01-01

**Authors:** Quansheng Sun, Tianjun Liao, Haibo Du, Yinfeng Zhao, Chih-Chiang Chen

**Affiliations:** 1School of Electrical Engineering and Automation, Hefei University of Technology, Hefei 230009, China; 2Academy of Military Sciences, Beijing 100850, China; 3Department of Systems and Naval Mechatronic Engineering, National Cheng Kung University, Tainan 70101, Taiwan

**Keywords:** genetic algorithm, LiDAR, MUAVs, optimization problem, map merging

## Abstract

The merging of environmental maps constructed by individual UAVs alone and the sharing of information are key to improving the efficiency of distributed multi-UAVexploration. This paper investigates the raster map-merging problem in the absence of a common reference coordinate system and the relative position information of UAVs, and proposes a raster map-merging method with a directed crossover multidimensional perturbation variational genetic algorithm (DCPGA). The algorithm uses an optimization function reflecting the degree of dissimilarity between the overlapping regions of two raster maps as the fitness function, with each possible rotation translation transformation corresponding to a chromosome, and the binary encoding of the coordinates as the gene string. The experimental results show that the algorithm could converge quickly and had a strong global search capability to search for the optimal overlap area of the two raster maps, thus achieving map merging.

## 1. Introduction

With the development of drone technology over recent years, unmanned aerial vehicles (UAVs) have been widely used in civil fields, such as power-line inspection, target tracking, agricultural plant protection, and geographic mapping [1,2]. However, in some cases, UAVs must have high environmental perception, such as for the task of search and rescue in disaster environments [3,4]. To achieve the autonomous planning and navigation of UAVs, the technique of collaborative simultaneous localization and mapping (SLAM) has become a popular topic for UAVs. A single UAV mapping in large scenes has many drawbacks, such as a small sensor range of action, limited observation angles, high computational complexity, weak storage capacity, and relying only on local sensor information. As a result, the localization errors for UAVs become increasingly larger, eventually leading to the drift phenomenon for mapping and localization. If multiple UAVs are employed in the environment, the whole system has stronger environmental detection capability and more accurate localization.

Raster maps are more advantageous in representing the real environment, and enabling precise positioning and navigation; research into the problem of merging together multirobot raster maps has attracted much attention [5,6,7,8]. There are two main types of merging maps. The first type is based on feature matching and mathematical optimization. The map-merging problem is modeled as an optimization problem [9,10]. The authors in [11] proposed an algorithm based on the Hough transform [12] to directly extract the local features of raster maps for stitching, which is fast, but the robustness of the algorithm is poor. In [13], the authors proposed the use of time-series-based algorithms for stitching images, but stitching using global iterative algorithms is inefficient. The other is a feature transformation algorithm based on scale invariability [14], and some improved algorithms based on SIFT were given [15,16]. The authors in [17] proposed image stitching using scale-invariant algorithms based on image alignment algorithms that are highly robust, but have high algorithmic complexity and slow speed, and are not conducive to the rapid creation of global maps. Traditional map-merging algorithms need to traverse all possible alignment points in all images, calculate the sum of squares of the corresponding pixel grayscale differences at each alignment point, and take the smallest one as the best alignment point. This results in a large number of tedious calculations and greatly increases the time redundancy of the algorithm. However, the GA-based map-merging method [18] can greatly reduce computational effort while maintaining matching accuracy.

In a two-dimensional raster map, there are three degrees of freedom representing translation in the *X* and *Y* axis directions, and a rotational angle θ. One raster map is fixed, and the other is rotated and translated. The aim is to find the planar transformation that corresponds to the optimal overlap between the two raster maps. In this way, raster map merging is transformed into a multidimensional combinatorial optimization problem. Genetic algorithms (GAs) are powerful for optimization problems [19,20] because of their ability to find global optima, and their high parallelism. In recent years, many researchers use genetic algorithms to solve some optimization problems in image processing [21]. We also performed research [22] on genetic algorithms in image processing. An improved algorithm was used to conduct template matching on a single image in this paper, but the real-time processing of multiple images is not involved. In [23], we applied the improved genetic algorithm to map the fusion of dual drones. Compared to these two papers, mutation improvements in genetic algorithm is taken for processing of large amounts of map data faster. There would be no overlap of local maps at all due to the distributed mapping–centralized merging method when the working area is large. A new method of merging maps based on DCPGA algorithm is proposed as shown in Figure 1.

In this case, the DCPGA algorithm still receives an erroneous optimal solution. A judgment function was added in this paper. We propose an improved genetic algorithm based on a directional cross and multidimensional perturbation mutation (DCPGA). The contributions of this paper are summarized as follows:The crossover and variation operators are improved compared with traditional genetic algorithms. A directed crossover operator and a multidimensional adaptive perturbation mutation operator are designed. To keep the population evolving excellently, a directional cross operator is employed. Individual variations in the population take different forms of variation depending on the fitness via the multidimensional adaptive variation operator.The raster maps are filtered on the basis of raster thresholds, and a mathematical model to optimize the merging maps is constructed. The DCPGA algorithm is applied to solve optimization problems in the process of merging maps, and achieves the real-time map merging of two UAVs with LiDAR. A distributed mapping–centralized merging method is used to improve the system’s efficiency, and a judgment function is introduced to avoid overlay between the two maps.

## 2. Analysis of Map-Merging Model

### 2.1. Problem Description

As shown in Figure 2, simultaneous localization and mapping (SLAM) is a typical mission for UAVs to perform navigation tasks. However, when a single UAV is used to explore large and complex environments, there are problems such as a small sensor range, limited observation capability, large amounts of data processing, high computational complexity, poor storage, and insufficient robustness and anti-interference capability. To this end, MUAV collaboration was employed in this paper to perform SLAM, and LiDAR sensors were used. To describe the process, the left subfigure shows that two LiDAR-equipped UAVs separately explore the environment and build maps. Then, they transmit the map information to the computer in real time, in which real-time map merging is performed to obtain a global environment map.

To achieve map merging, a reference coordinate is required to be determined at the first step; usually, one map is chosen as the reference map. The next step is to find the optimal matching matrix T={R,t} of the two maps. *R* is a two-dimensional rotational matrix related to rotational angle θ, and *t* is a two-dimensional column vector, as shown in Equation (Equation 1).
(1)$$R=\left[\begin{array}{cc} \cos\theta & -\sin\theta \\ \sin\theta & \cos\theta \end{array}\right],t=\left[\begin{array}{c} t_{x}\\ t_{y} \end{array}\right].$$

As shown in Figure 3, there are two grid maps named $M_{A}$ and $M_{B}$ built by two UAVs. Assume that there is some overlap between these two maps. $M_{A}$ was set to be the reference map. The purpose of map merging is to find an optimal rotation and translational matrix T={R,t} with the maximal matching degree of the overlapping region of maps $M_{A}$ and $M_{B}$. Once matrix *T* is obtained, a global map $G_{M}$ is produced.

### 2.2. Mathematical Mode

Figure 4 shows the source maps that contain environmental obstacles, free space, and noise data. Hence, it is necessary to filter the initial map data, and a method of setting the pixel threshold is used. In this paper, the threshold range of free space on a raster map was chosen to be 0–50, the range of the obstacle space on the map was chosen to be 51–100, and the threshold of the unknown environment was −1. On the basis of these settings, the obstacle map data were extracted, as shown in Figure 5.

Obstacle point cloud maps were assumed to be $M_{A}$ and $M_{B}$. The objective of map merging is to maximize the overlap between $M_{A}$ and $M_{B}$. To match two maps directly, the following assumption is proposed. In an obstacle map, obstacles are equally likely to exist around each pixel point. If the point is closer to the obstacle point, the likelihood of an obstacle existing is higher. On the basis of this assumption, we calculated Euclidean distance matrix ${\bar{M}}_{A}$ for map $M_{A}$, where the pixel value of each point was the distance from the current point to the nearest obstacle point. It is mathematically described as follows.
(2)$$f_{dist}\left({\bar{M}}_{A}\left[x,\;y\right]\right)=\min_{\left(x_{i},y_{i}\right)\in M_{A}}\left(\sqrt{{\left(x-x_{i}\right)}^{2}+{\left(y-y_{i}\right)}^{2}}\right),$$
where (x, y) and $\left(x_{i},\;y_{i}\right)$ are each point coordinate for matrices ${\bar{M}}_{A}$ and $M_{A}$, $f_{dist}\left({\bar{M}}_{A}\left[x,\;y\right]\right)$ is the value for each element in matrix ${\bar{M}}_{A}$ corresponding to coordinate (x, y). After the calculation, the map corresponding to matrix ${\bar{M}}_{A}$ is shown in Figure 6. The sparse point set was clearly transformed into a dense point set.

To achieve map merging, let ${\bar{M}}_{A}$ be the reference map, and perform a pose transformation for $M_{B}$ on the basis of Relation (Equation 3). Rotational transformation is *R*, and translational transformation is *t*, which is shown in (Equation 3):(3)$${\bar{M}}_{B}\left[x,\;y\right]=\underset{R}{\underset{︸}{\left(\begin{array}{cc} \cos\theta & -\sin\theta \\ \sin\theta & \cos\theta \end{array}\right)}}M_{B}\left[x,\;y\right]+\underset{t}{\underset{︸}{\left(\begin{array}{c} t_{x}\\ t_{y} \end{array}\right)}}.$$

On the basis of transformed matrix ${\bar{M}}_{B}$, the map-merging problem is translated into multiplying the overlapping areas of ${\bar{M}}_{A}$ and ${\bar{M}}_{B}$ to find the minimal value. The optimal problem is described as follows:(4)$${\arg\min}_{\left(\theta ,t\right)}\left[\sum_{(x,y)\in \Omega }\;\;\;\left({\bar{M}}_{B[x,y]}\right)\;\cdot \left({\bar{M}}_{A[x,y]}\right)\right],$$
where set Ω is the points that are the overlapping part between matrices ${\bar{M}}_{A}$ and ${\bar{M}}_{B}$. In this way, merging raster maps is seen as a multidimensional combinatorial optimization problem where one raster map is fixed, and the other is rotated and translated with the aim of finding the planar transformation corresponding to the optimal overlap of the two raster maps. To solve this optimization problem, this paper proposes an improved genetic algorithm.

## 3. Improved Genetic Algorithm

Genetic algorithm (GA) [24,25] is an adaptive heuristic search algorithm inspired by the process of evolution. It is a population-based optimization technique. In this paper, we use the floating-point encoding method. The construction of a GA for a point-matching problem involves the following steps: selection, crossover, and mutation. The proposed DCPGA incorporates operations of sorting and classification on chromosomes before crossover and mutation. The global optimization problem is considered as follows:(5)minF(x),x⊂X,
where *x* is a continuous variable with domain $X\subset R^{D}$ defined by each element satisfying constraint [l, u] and $R^{D}$ is the *D*-dimensional solution space. Function F(x): X→R is a continuous real-valued function. A point $x^{*}\in X$ is called the global minimizer of *f* if $F\left(x^{*}\right)\leq F\left(x\right),\forall x\in X$. GAs maintain a set of potential solutions $X_{a}=\left\{x_{1},\cdots ,x_{n}\right\}$ at every time step *a*, called a generation. At each generation *a*, the DCPGA performs selection, crossover, and mutation to update the current population of solutions $X_{a}$. If the solution is multidimensional, one potential solution is defined as $x_{a}={\left[x_{a,1},\ldots ,x_{a,D}\right]}^{T}$.

In a two-dimensional raster map, there are three degrees of freedom representing the displacement and deflection angles in the *X* and *Y* axis directions, respectively. The position of the chromosome is, therefore, coded as $x_{i}=[\begin{array}{ccc} t_{x} & t_{y} & \theta {]}^{T} \end{array}$.

### 3.1. Initial Population

Initially, $X_{a}$ is generated randomly between the upper and the lower boundaries of the solution space. Next, $X_{a}$ is created from the genetic encodings of the fittest parents via crossover operators, and new points in the solution space are sampled using mutation operators. We used a roulette wheel to select all possible solutions. The fitness of chromosome $x_{i}$ was $f(x_{i})$, and the number of populations was *n*. The probability of it being retained is calculated according to Equation (Equation 6):(6)$$p\left(x_{i}\right)=\frac{f(x_{i})}{\sum_{k=1}^{n}f(x_{k})}.$$

This method of selection ensures that all individuals have a certain chance of being retained. If the fitness of an individual is greater, it has a greater chance of being retained.

### 3.2. Directional Crossover

In a series of genetic operators, the crossover operator generates better individuals through gene pairing and crossover. The main operation of the crossover operator is to achieve genetic optimization. Before crossover operation, we sorted the chromosomes as $X_{a}=\left\{{x_{a}}^{1},{x_{a}}^{2},\ldots ,{x_{a}}^{d}\right\}$ and chose the excellent part according to their fitness. In each generation, we set the excellent individuals as the parents, and the remaining chromosomes of the population were crossed with them at the same dimension according to Equation (Equation 7):(7)$${\bar{x}}_{a,j}^{i}=x_{a,j}^{i}+\frac{f_{d}}{f_{i}+f_{d}}\times \left(x_{a,j}^{d}-x_{a,j}^{i}\right).$$

In Equation (Equation 7), ${\bar{x}}_{a,j}$ represents the *j*-th dimension chromosome generated by the crossover, $x_{a,j}^{i}$ represents the least adaptive chromosome in the *a*-th generation, $x_{a,j}^{d}$ represents the excellent chromosome in this generation, and $f_{i},f_{d}$ represent the fitness of individuals $x_{a,j}^{i}$ and $x_{a,j}^{d}$. During the crossover, traits inherited by the offspring are influenced by parental adaptation, and the next generation could receive better features from parents. The convergence speed is improved by this phenomenon. The directional crossover process is shown in Figure 7.

### 3.3. Multidimensional Perturbation Mutation

Genetic optimization algorithms tend to fall into the local optimum and cause the convergence result to deviate greatly from the optimal solution. In order to improve population diversity, and reduce the degree of population aggregation, mutational operations are introduced. When the algorithm has a precocious convergence phenomenon, the optimal position of the group is a locally optimal solution. The individuals tend to move closer to the optimal position of the group, at which time the individuals gather near the local optimal solution and lose the ability to find a better solution existing in other positions. If the optimal position of the group can be perturbed out of the local optimal solution, the algorithm would achieve further convergence. Therefore, globally optimal individuals and the method of multidimensional random perturbation of the optimal position of the group were mutated to jump out of the local optimal solution in this paper. The variation method is as follows.

It was assumed that the p% of the optimal position in dimension *D* of the population selected for random perturbation and the *j*-th dimension random perturbation were given as shown in Equation (Equation 8).
(8)$$x_{a+1,j}^{i}=x_{a,j}^{i}\times \left(1+A\cdot randn\left(\right)\right).$$
(9)$$A=\frac{1}{2}\left(1-\frac{t}{C}\right).$$
where *A* is the perturbed amplitude that decreases with the iterations, and randn() is a random variable that follows a standard normal distribution.

The mutation probability of the heuristic optimization algorithm should be adjusted with reference to the population aggregation degree. If the population size is *N* and at the *t* iteration, the coordinate position vector of each individual is expressed as $X_{i}^{t}$, and σ is the population standard deviation, then σ can be defined as (Equation 10):(10)$$\sigma =\frac{1}{\sigma_{0}}\sqrt{\sum_{i=1}^{N}{∥X_{i}^{a}-X_{avg}^{a}∥}^{2}},$$
where $\sigma_{0}$ is the normalization factor. It is the standard deviation of the population that is not normalized at the time of population initialization. $X_{avg}^{t}$ is the centroid of the population, calculated with Equation (Equation 11).
(11)$$X_{avg}^{a}=\frac{1}{N}\sum_{i=1}^{N}X_{i}^{a}.$$

In the previous experiments, the population standard deviation was concentrated in the interval [0, 1]. When the population standard deviation is close to 0, the population is denser, and a large variation probability should be adopted. If the standard deviation of the population is close to 1, the population is dispersed, and a small variation probability should be adopted. This property is conducive to establishing the probability of variation as a function of the standard deviation of the population. When the standard deviation of the population is close to 1, it indicates that the population has received the local optimal solution, and a larger variation probability should be used. Conversely, when the standard deviation of the population is close to 0, the lower probability of variation is used.

Therefore, the variation probability calculation formula is established on the basis of population standard deviation and iteration times as in (Equation 12).
(12)$$p=\omega_{p\sigma }\left(1-\sigma \right)-\omega_{pt}\frac{a}{C}+b.$$
where $\omega_{p\sigma }$ is the weight of the standard deviation of the mutational probability. $\omega_{pt}$ is the weight of the iterations of the mutational probability. *b* is the offset of the variation probability and it is a constant. *C* is the maximal number of algorithmic iterations. $\omega_{p\sigma }$, $\omega_{pt}$, and *b* are parameters to be set and we experimented several times to find the suitable values of $\omega_{p\sigma }=1.5$, $\omega_{pt}=1.0$ and b=0.

### 3.4. Judgment Function

The map-merging function is solved on the basis of the assumption that there are overlapping areas between local maps. There is no overlap of local maps at all due to the distributed mapping–centralized merging method when the working area is large. In this case, the DCPGA algorithm still receives an erroneous optimal solution. Therefore, a judgment function is added to determine whether the maps were merged successfully. The judgment function is given by combining the raster similarity and confidence *c* of overlapping areas as follows:(13)$$jud\left(M_{A},M_{B}\right)=\frac{com(M_{A},M_{B})}{com\left(M_{A},M_{B}\right)+dif\left(M_{A},M_{B}\right)}.$$

In (Equation 13), $com(M_{A},M_{B})$ and $dif(M_{A},M_{B})$ are the number of rasters with the same probability and the number of rasters with different probabilities in converted maps $M_{A}$ and $M_{B}$. They are calculated as in Equations (Equation 14) and (Equation 15).
(14)$$sam\left(M_{A},M_{B}\right)=num\left\{q=\left(x,\;y\right)|M_{A}\left[q\right]=M_{B}\left[q\right]\in c\right\}.$$
(15)$$dif\left(M_{A},M_{B}\right)=num\left\{q=\left(x,\;y\right)|M_{A}\left[q\right]\neq M_{B}\left[q\right]\in c\right\}.$$

Obviously, if the overlap of maps is larger, the value of $jud(M_{A},M_{B})$ is larger. In the experiments, when the value of $jud(M_{A},M_{B})>0.8$, it was considered that the maps were merged successfully.

## 4. Experiments and Analysis

Four test functions were chosen to verify the effectiveness of the DCPGA, and three genetic algorithms, IGA [26], IRCGA [27], and TAGA [28], were selected in the experiments to compare the algorithms’ accuracy and convergence speed. Iteration times and the optimal solution were chosen as the termination criteria. The relevant experimental results were proven in previous work.

In the tests, the population size of each algorithm was set to 200, the number of iterations was 100, and each algorithm ran independently for 30 times. The four tested algorithms were as follows:

DCPGA: the directional cross-genetic algorithm proposed in this paper; the algorithmic parameters were set as follows: $p\%=10\%,\omega_{p\sigma }=1.5,\omega_{pt}=1.0,b=0,C=100,D=40$.

IGA: an improved genetic algorithm based on real number coding that is a classical algorithm that achieves adaptive adjustment during operation by changing the crossover and mutation probability: mutational parameter r=0.3,b=2; adaptive parameters $k_{1}=k_{2}=1.5,k_{3}=k_{4}=0.5$.

IRCGA: an improved real-coded genetic algorithm that adopts the strategy of adaptively changing $p_{c}$ and $p_{m}$, and performs multiparental crossover: $p_{1}=0.55,p_{2}=0.3$, $c_{1}=c_{2}=1.0$.

TAGA: an optimized genetic algorithm that learns from the best individual: $p_{c}=0.55$, $p_{m}=0.3,a_{i}\in \left(-0.5,\;1.5\right),c_{1}=c_{2}=1.5$.

We chose the following four test functions:

Bohachevsky function: a unimodal bowl function that slowly converges to minimal value 0 at [0,0].
(16)$$f_{1}=0.7+x^{2}+2y^{2}-0.3\cos\left(3\pi x\right)-0.4\cos\left(4\pi y\right).$$

Sum-squares function: a unimodal function that obtains minimal value 0 at [0,0].
(17)$$f_{2}=x^{2}+2y^{2}.$$

Griewank function: a multimodal function that obtains minimal value 0 at [0,0].
(18)$$f_{3}=\frac{x^{2}+y^{2}}{4000}-\left[\cos\left(x\right)+1\right]\times \left[\cos\left(\frac{y}{\sqrt{2}}\right)+1\right].$$

DropWave function: a multimodal function that obtains the minimal value of −1 at [0,0].
(19)$$f_{4}=-\frac{1+\cos(12\sqrt{x^{2}+y^{2}})}{0.5(x^{2}+y^{2})+2}.$$

### 4.1. Simulation Results and Analysis

The standard deviation of the optimal adaptation value and the running time of the algorithm of completing 100 iterations were the main comparison data in this paper. The results of the tests are shown in Table 1 and Table 2.

In Table 1, $f_{1}$ and $f_{2}$ are single-peaked functions with relatively low optimization difficulty. They could better reflect the local fine search capability of the algorithm. $f_{3}$ and $f_{4}$ are multimodal functions that are more likely to trap the optimization algorithm in a local mean. Combined with the information in Table 2, the DCPGA proposed in this paper was less likely to fall into a local optimum solution, and could significantly improve the convergence speed of the algorithm.

### 4.2. Two UAV Map-Merging Experiment

Before the experiments, we tested the wireless communication function between the UAVs and the computer to ensure that the LiDARs could be switched on and off remotely by the computer during the experiments. All the communication was handled by a robot operating system (ROS) [29], that was capable of transmitting specific data structures in the scope of robotics. We recorded data packets on the basis of the ROS. The experimental environment was a room with an area of 72 square meters as shown in Figure 8. Due to the indoor environment, the drone flew at an altitude of around 2 m. A TOF laser rangefinder was installed on the drone to ensure that the drone flew as high as possible at around 2 m. During the experiments, two UAVs equipped with S1 LiDARs and an NVIDIA Jetson Xavier NX were controlled with a remote control to fly. The flight trails of two UAVs are shown in Figure 9.

The establishment results of the submaps and the fusion maps are shown in Figure 10. There was strong similarity between $M_{A}$ and $M_{B}$ in the 65th frame. However, due to the existence of the judgment function, it was not able to merge them into a map. In the 125th frame, a common part also appeared in $M_{A}$ and $M_{B}$ when the two drones flew to the same area. The fusion map was more reasonable. In the 175th frame, although submaps $M_{A}$ and $M_{B}$ were not completed, the main control computer used the overlapping information of the two drones to obtain the overall situation of the environment, and it was consistent with the final surveying and mapping results of $M_{A}$ and $M_{B}$. Without other positioning methods, this design only uses data from LiDARs to complete the grid map merging.

We experimented with a single UAV 10 times and recorded the time of each experiment. For comparison, the experiments of mapping with two UAVS were carried 10 times. The results are shown in Table 3, which shows that the proposed algorithm could effectively reduce the required time for mapping.

## 5. Conclusions

This paper focused on the problem of raster map merging when multiple UAVs are used for collaborative exploration. The theory of image matching was adopted in this paper. A mathematical model was constructed to transform the raster map-merging problem into a multidimensional combinatorial optimization problem. A DCPGA was proposed based on the random distribution of the optimization objective function. Directional crossover kept the algorithm searching in the direction of the optimal solution. Multidimensional perturbation variants could overcome the prematureness problem of genetic algorithms, and reduce the possibility of genetic algorithms falling into local optimum solutions. Experimental results from algorithmic simulations showed that the DCPGA had a fast convergence rate and strong global convergence capability. In the experiments of merging maps, the raster map similarity was used as a metric, and the Euclidean distance transform and the improved algorithm performed a global search for the best overlap between each local map. The experimental results demonstrated the feasibility and effectiveness of the method in complex environments.

The proposed method was only applied to offline map merging during exploration experiments with two UAVs. Research will next be conducted to achieve the online real-time map merging of two drones, and the collaborative mapping of more drones. The DCGPA will also be improved to improve the efficiency and accuracy of mapping.

## Figures and Tables

**Figure 1 sensors-23-00447-f001:**
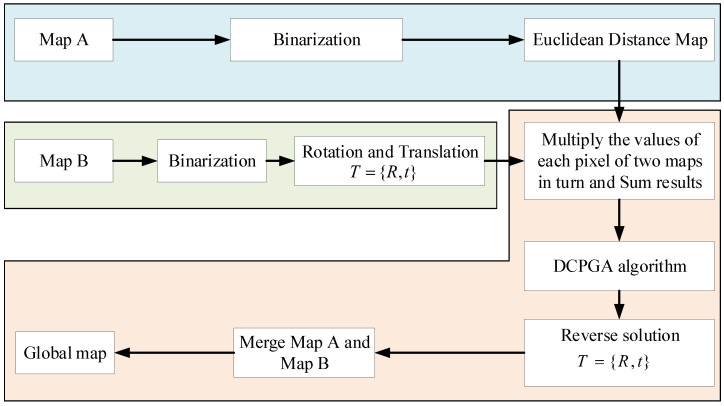
Map-merging frame based on DCPGA algorithm.

**Figure 2 sensors-23-00447-f002:**
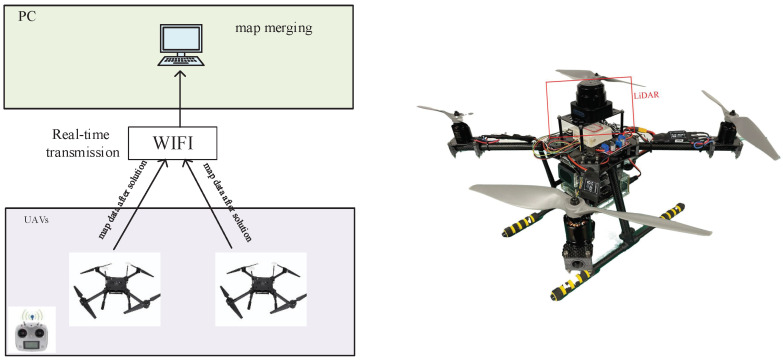
MUAV collaboration.

**Figure 3 sensors-23-00447-f003:**
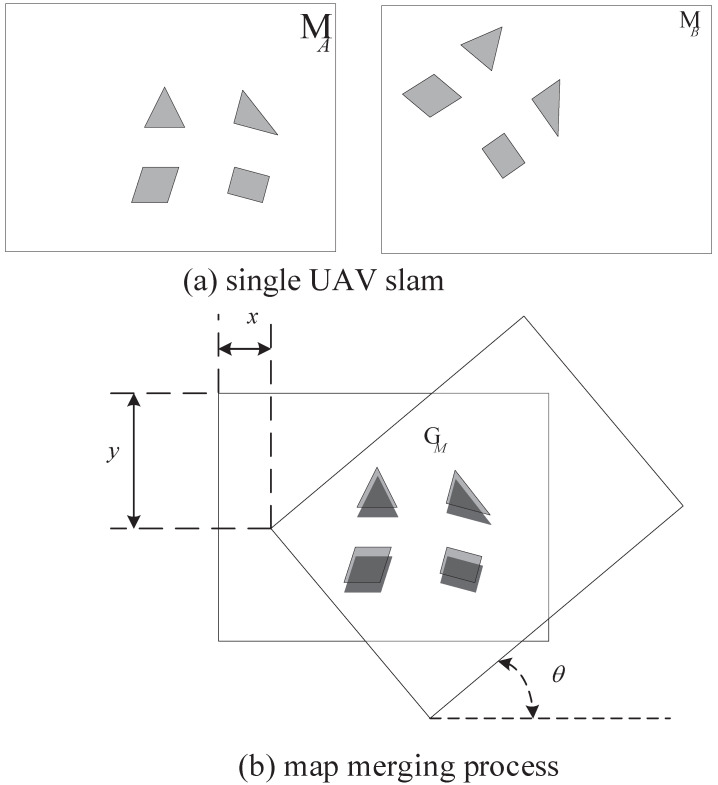
Map-stitching diagram.

**Figure 4 sensors-23-00447-f004:**
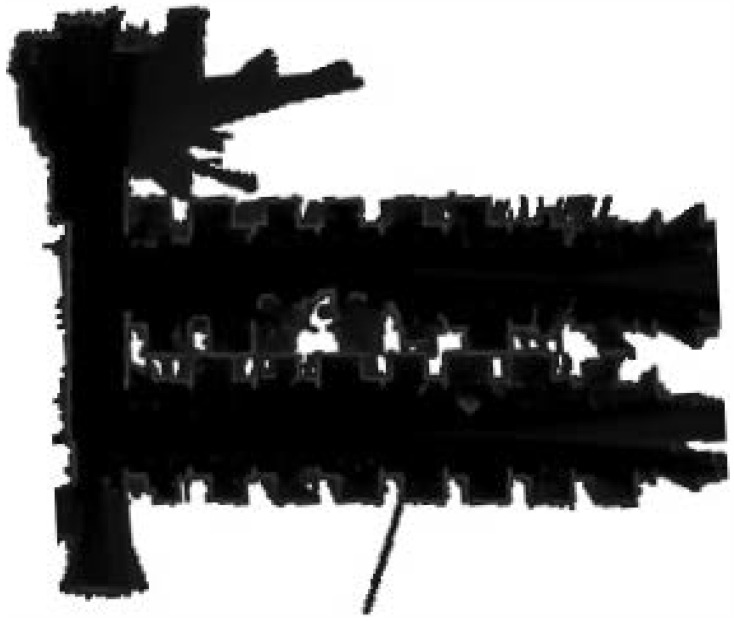
Map with source data.

**Figure 5 sensors-23-00447-f005:**
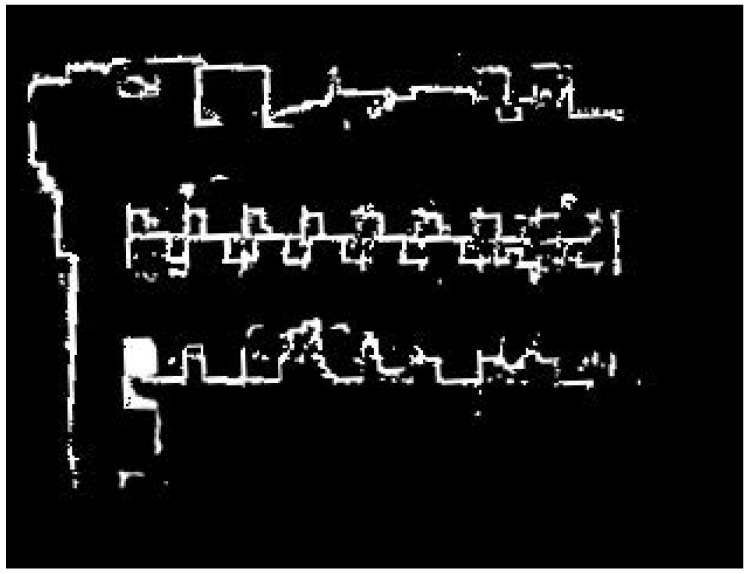
Map with filtered data.

**Figure 6 sensors-23-00447-f006:**
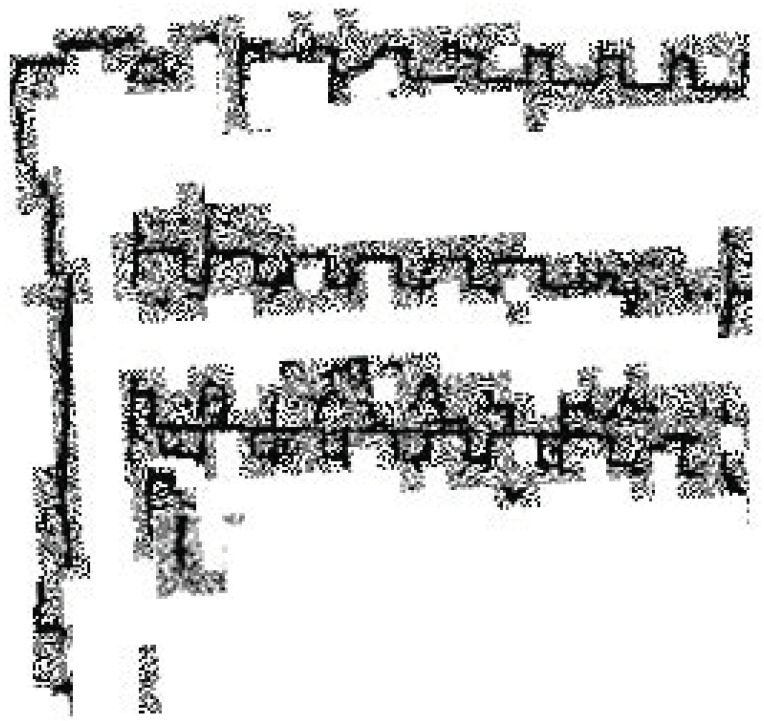
The map corresponding to Euclidean distance matrix ${\bar{M}}_{A}$.

**Figure 7 sensors-23-00447-f007:**
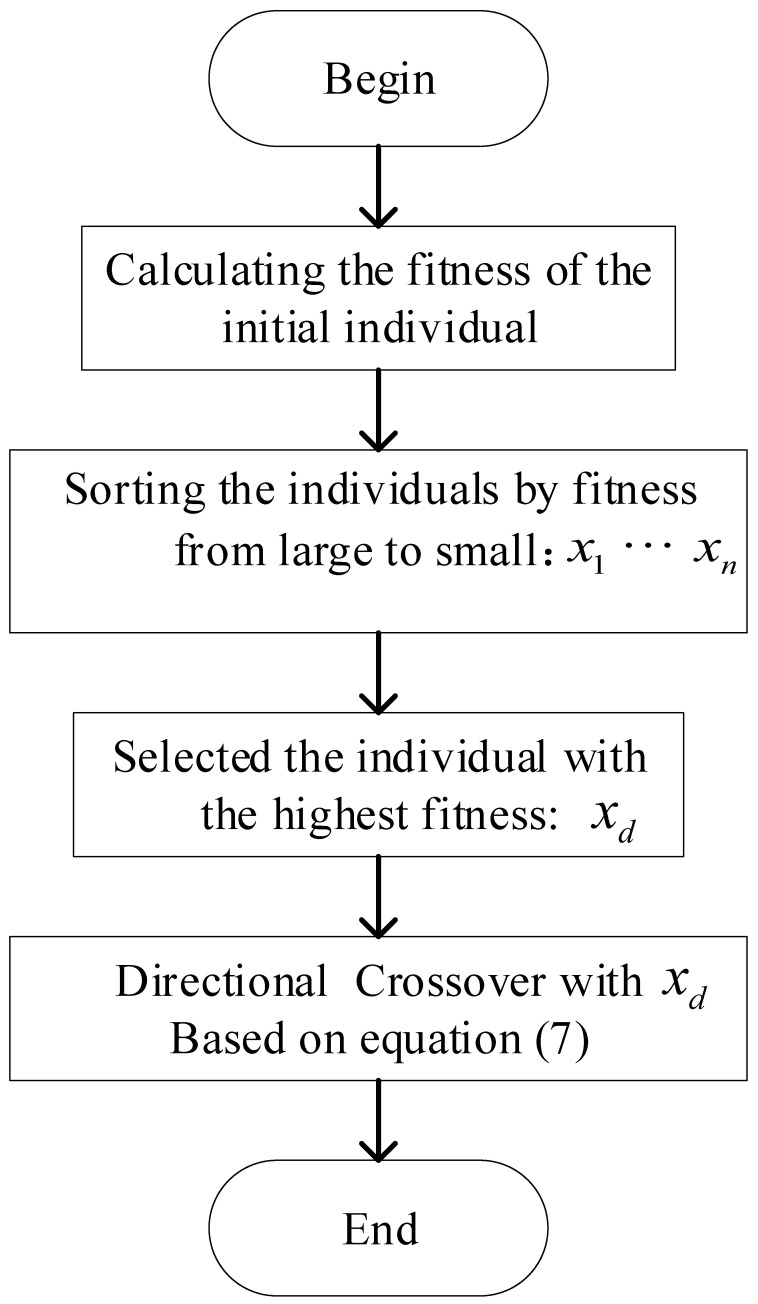
Schematic diagram of directional crossover process.

**Figure 8 sensors-23-00447-f008:**
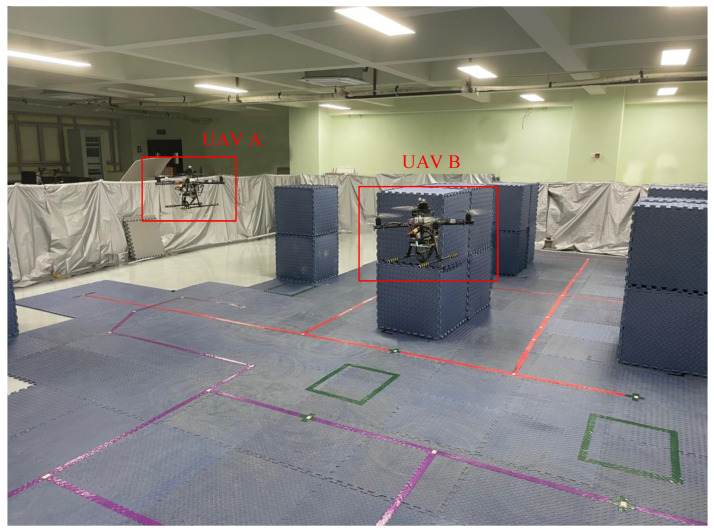
Experimental environment.

**Figure 9 sensors-23-00447-f009:**
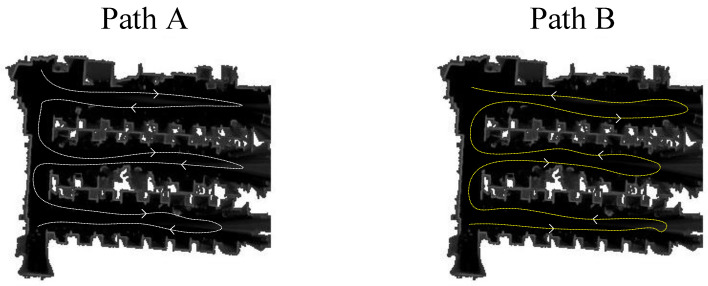
Paths of UAVs.

**Figure 10 sensors-23-00447-f010:**
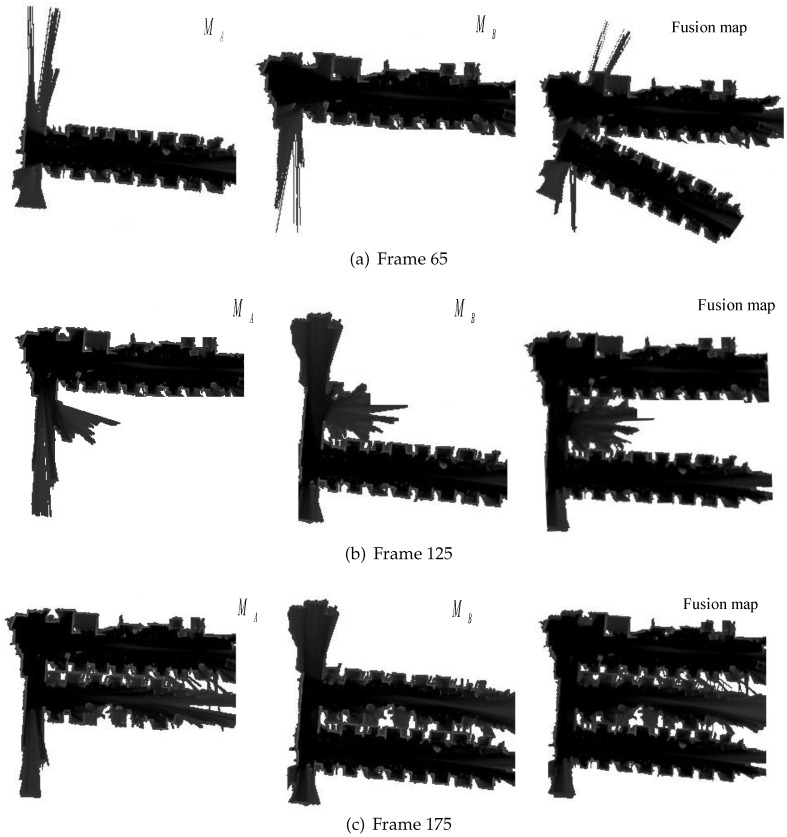
Map-merging process.

**Table 1 sensors-23-00447-t001:** Standard deviation comparison.

	Standard Deviation	DCPGA	IRGA	IGA	TAGA
Functions					
$f_{1}$		2.32 × 10${}^{-3}$	4.69 × 10${}^{-3}$	3.17 × 10${}^{-3}$	2.37 × 10${}^{-3}$
$f_{2}$		3.51 × 10${}^{-4}$	4.27 × 10${}^{-3}$	3.46 × 10${}^{-3}$	2.38 × 10${}^{-3}$
$f_{3}$		2.84 × 10${}^{-3}$	1.48 × 10${}^{-2}$	2.83 × 10${}^{-2}$	2.15 × 10${}^{-2}$
$f_{4}$		1.93 × 10${}^{-2}$	8.04 × 10${}^{-2}$	4.69 × 10${}^{-1}$	1.73 × 10${}^{-1}$

**Table 2 sensors-23-00447-t002:** Time comparison.

	Time(s)	DCPGA	IRGA	IGA	TAGA
Functions					
$f_{1}$		1.19	1.64	3.83	1.35
$f_{2}$		1.34	1.36	4.34	1.74
$f_{3}$		1.66	1.79	3.93	1.59
$f_{4}$		2.19	2.24	4.19	2.35

**Table 3 sensors-23-00447-t003:** Mapping time comparison of a single UAV vs. MUAVs.

UAV	Numbers	Minimal Time (s)	Maximal Time (s)	Average Time (s)
Single UAV	10	118	470	313
MUAVs	10	97	185	132

## Data Availability

Not applicable.

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
