# Peer review of "A Method of Merging Maps for MUAVs Based on an Improved Genetic Algorithm"

_sensors, 2023, doi:10.3390/s23010447_

Round 1
Reviewer 1 Report
An improved genetic algorithm is proposed in this paper to deal with the real-time map merging problem for multiple unmanned aerial vehicles (MUAVs). This is an interesting research topic. However, the description of the map-merging method based on DCPGA is not detailed enough. Some comments are given as follows that may be helpful to improve the quality of the paper.
· It is suggested to give a detailed flowchart of how to solve the optimization problem in the process of merging maps based on DCPGA .
· The problem of the same variable symbol representing different meanings exists in the paper. For instance, the matching matrix T ={R, t} of two maps in Page3 is different from the T in equation (11); and t=(tx,ty) in equation (3) is different from the t in equation (11).
· ‘’D is the distance of the chromosome in the solution space ‘’in Figure 6, while D represents the dimension of the solution space in the last line of page 5. Do these two expressions mean the same thing?
· Please explain the meaning of the expression [f(xi)+f(xd)]×D in Figure 6.
· I think the function f(¬) in equation (5) and that in equation (6) represent different meanings and should not be expressed with the same symbol. The smaller the value of the objective function, the greater the fitness.
· It is difficult to understand the first sentence on page 7, “It is assumed that p of the optimal position in dimension D of population selected for random perturbation and the j-th dimension random perturbation is given equation (8).”
· What does the perturbation amplitude A in equation (8) depend on? Is there a general range for the value of A?
· What does T represent in equation (11)? How to determine the values of two weights and offset b? Does p in equation (11) represent probability of variation? If yes, it may be different from the variable p in equation (13) and (14), so it is suggested to use different symbols to avoid ambiguity.
· There are some incorrect English expressions in the first paragraph of Section 4, such as ‘‘tested function are choosed to verify the effectiveness of the DCPGA algorithm’’ and ’’comparison of algorithms’ the accuracy and convergence speed’’.
· What is the solution space of the map merging optimization problem? It is not very clear in the paper. See the sentence in Section 4.1, ‘’In the map merging problem, the chromosomes xi corresponds to the pixel coordinates’’; and the sentence in Section 5,‘’The multi-dimensional adaptive mutation directional cross-genetic algorithm is used for global optimization, and the optimal transformation matrix is quickly and efficiently searched.’’
· In Section 4, it is suggested to give the key parameters of DCPGA in the multiple UAVS map-merging experiment, such as the population size and the parameters associated with crossover and mutation operations.
Reviewer 2 Report
The experimental part is completely missing. It needs more explanation and clarity
DCPGA algorithm, and other three genetic algorithms IGA [23], IRCGA [23], and TAGA comparison is not given. Please make a table and compare them for accuracy and convergence speed.
UAV flight trails are not shown. What altitude it was flown. How you have optimized the data. It needs more explanation
Explain in detail about collection of point cloud LiDAR data with pre and post processing steps
What are the on-board capabilities and how you have handled the LiDAR data.
What is the use of genetic algorithm? How it was effectively addressed the present problem. Discussions to be added.
Abstract and conclusion should be improved
Need rigorous literature review with recent papers.
Round 2
Reviewer 2 Report
Still it is not clear the map merging process. Please include the clear picture of your map that is considered. The maps are not clear.
Still you have to show the two UAVs flying photographs and experimental section to be improved.
Please dont mention table word in the caption
Conclusion needs to be improved with scientific merits achieved
Sentence formation is not good in the conclusion
Round 3
Reviewer 2 Report
"The 10 experiments.... ". It is not the way to write it.
Please include the UAV photograph with the map in the manuscript as you have given in the response file.
And we constructs a mathematical model. In the conclusion. Please write properly. "And" is not the way to start a sentence
Experimental results from algorithm simulations show that The algorithm.
Correct the above sentence
